# Beach Scenic Quality versus Beach Concessions: Case Studies from Southern Italy

Alexis Mooser [1,2,*], Giorgio Anfuso [2], Enzo Pranzini [3], Angela Rizzo [4] and Pietro P. C. Aucelli [1]

1   Department of Science and Technology (DiST), Parthenope University, 80143 Naples, Italy
2   Faculty of Marine and Environmental Sciences, University of Cádiz, 11510 Cádiz, Spain
3   Department of Earth Sciences, University of Florence, 50121 Florence, Italy
4   Department of Earth Sciences and Geo-Environmental (DISTeGEO), University of Bari Aldo Moro, 70122 Bari, Italy
*   Correspondence: alexis.mooser@alum.uca.es

**Abstract:** This paper essentially aims to identify coastal sites of great scenic value not (or barely) affected by human intrusions and propose sound management interventions to improve their landscape quality. Today, management of coastal areas in Italy is a very complex task essentially because of institutional fragmentation and overlapping of laws/regulations at the national, regional and municipal levels. It is estimated that only half of the country's beaches are freely accessible and usable for bathing, i.e., 43% are occupied by private concessions and in 7.2% bathing is not allowed because of water pollution. Sites' scenic quality was assessed using the Coastal Scenic Evaluation System (CSES), a robust semi-quantitative methodology based on a set of 26 physical/human parameters, weighting matrices parameters and fuzzy logic mathematics. An evaluation index (D) was afterward obtained for each site and used to classify them into five scenic classes. After a long process of field testing along the coasts of the Tyrrhenian, Ionic and Adriatic seas (25 municipalities, 7 provinces and 4 regions: Campania, Basilicata, Calabria and Apulia), a total of 36 sites were selected for this paper. Twenty-four sites fall within Class I, i.e., were extremely attractive (D ≥ 0.85; CSES) because of their exceptional geomorphological settings that favour a wide variety of coastal sceneries. Most of Class II (8) and Class III (4) sites could be upgraded by implementing clean-up operations or by reducing intrusive beach facilities. Meanwhile "private" beaches are usually cleaned; beach litter at remote/public sites represents a big concern to be challenged. Today, finding a free/aesthetic/clean beach without human intrusions in a fully natural environment is far more complex than it seems. Given this context, emphasis was particularly placed on beach litter and concessions aspects.

**Keywords:** CSES; beach management; litter; concessions; protected areas; national parks; 3S tourism; Campania; Basilicata; Calabria; Apulia

## 1. Introduction

Sea bathing was discovered between the 18th and 19th centuries in northern European countries where the present way of enjoying a seaside holiday is less favoured because of weather conditions. In such countries, the pleasure to stay on the strand and dive into the sea became possible once people lost their ancestral fear and terror of beaches [1]. Moving to the coast and bathing was also stimulated by the persuasion of the health benefits that such activities could give. In fact, sea bathing was particularly discovered by the British as a form of therapy and Dr. Robert Wittie wrote the first publication that praised cold sea water as a medicine against various ailments, including gout and worms [2].

The British soon arrived on the French coast from where, thanks also to the development of the railway network, reached Liguria, and then, the Italian coast [3], perhaps mixing the grand tour with the "sun, sea and sand" (3S) tourism. It must be stressed that it is because of the emulation of these northern visitors that Italians started to go to the beach for leisure.

As early as in the 19th century some honeypots emerged, such as San Remo (Liguria), Viareggio (Tuscany), Taormina (Sicily) and Capri (Campania), further advertised by the cinema in years in which Italian films were seen all over the world [4]. Only the wealthy classes went on holiday to the seaside; however, towards the end of the 19th century and at the beginning of the 20th, this fashion spread to an increasingly vast bourgeoisie. It was with fascism that the activities in the open air had a great development in Italy and the "sea trains" allowed the inhabitants of the cities to spend summer Sundays at the sea [5]. This is how the conditions were created for that 'race to the sea' that characterised the years after the Second World War, when the economic boom enabled a substantial number of Italian families to afford holidays by the sea, thanks also to the recognition of paid leaves. Coastal settlements had a fast and unregulated expansion, very frequently with unauthorized constructions, which transformed many parts of the pristine Italian coast into linear cities that developed irrationally without urban planning [6]. Further, due to the geomorphological configuration of the Italian peninsula, with the Apennine Mountains frequently arriving to the sea, roads and railways were built along the coast, frequently on the beach itself.

Bathing services, from the ancient bathing machines evolved into huts on piers to which pubs and restaurants were added, and lines of cabanas grew alongshore at the foot of the dunes, or even on the top of them. The first real bathing establishments were built in northern Italy in Livorno (1781), Viareggio (1827) and Rimini (1843) [7], and nowadays, Italy is one of the most popular 3S destinations in the world, and surely the first in terms of exclusive beach clubs (*lido in Italian*), renting gazebos for up to 1000 EUR/day [8]. Even if the term *lido* (*lidi in plural*) is usually associated with beach clubs, it does not have a direct translation in other languages as it covers a wide variety of private beach complexes, from modest restaurants to extensive luxury clubs. Today, 42.8% of Italian beaches are assigned under concession to private persons or enterprises that occupy the strand with sun umbrellas, sunbeds, gazebos, etc., making access to the sea practically impossible for those who are unwilling or unable to pay the rent. Such concessions reach ca. 70% in the regions of Campania, Emilia-Romagna and Liguria. Even more outrageous, several municipalities achieve extreme rates such as Getteo in Emilia-Romagna (100% under concession), Pietrasanta (98.8%) and Camaiore (98.4%) in Tuscany [9]. Consequently, the use of beaches for leisure transformed coastal areas into places of strong economic interest and highly productive spaces [10], but their original environmental and landscape value was greatly reduced, and visitors looking for pristine environments had to move away from inhabited centers and abandon paved roads. Another very recurrent problem in Italy is the lack of beach management at "free" and rural/remote sites (without concessions) that can be linked to low interest from local managers [9]. The usual presence of large amounts of beach litter is quite symptomatic of this situation. An alarming report concerning various Italian regions, revealed that, on average, on Italian beaches 10 litter items per square meter are recorded, most of them (>80%) consisting of plastic materials [11].

Nonetheless, it is self-evident that countries with superb coastal landscapes have an invaluable attraction for tourism. As stated by Williams [12], scenic quality is among the five parameters of the greatest significance for coastal tourists, i.e., with safety, facilities, water quality and litter. The aesthetic of sceneries may be perceived as a combination between "attractiveness" and "integrity"; whilst "attractiveness" measures/considers the intrinsic beauty of a landscape (based on human perceptions), and "integrity" indicates its degree of intactness and wholeness [13]. Based on this, scenery is not only a socio-environmental (non-renewable) resource but also an economic one, and must be regarded as a strategic issue to be faced. In line with the European Landscape Convention [14], it is mandatory to soundly manage/preserve the remaining natural sites of great aesthetic value in which tourists are interested, and scenic assessment constitutes for that purpose, an outstanding tool for coastal managers/planners. Given this background, this paper aims to identify top scenic sites with low human intrusion, quantify their attractiveness through the CSES method (further described) [15–17], and point out ways to improve/recover their

scenic beauty. During the field work observations, emphasis was given to coastal scenic quality (principally) and on beach litter, private concessions and associated adverse effects.

## 2. Study Area

The 36 investigated coastal sites are located in the southern Italian provinces of Naples and Salerno (Campania Region), Potenza and Matera (Basilicata), Cosenza (Calabria) and Foggia and Lecce (Apulia Region) that stand out for showing a wide range of morphological systems characterised by alternating low-lying and high coasts along the Tyrrhenian, Ionian and Adriatic seas (Figure 1).

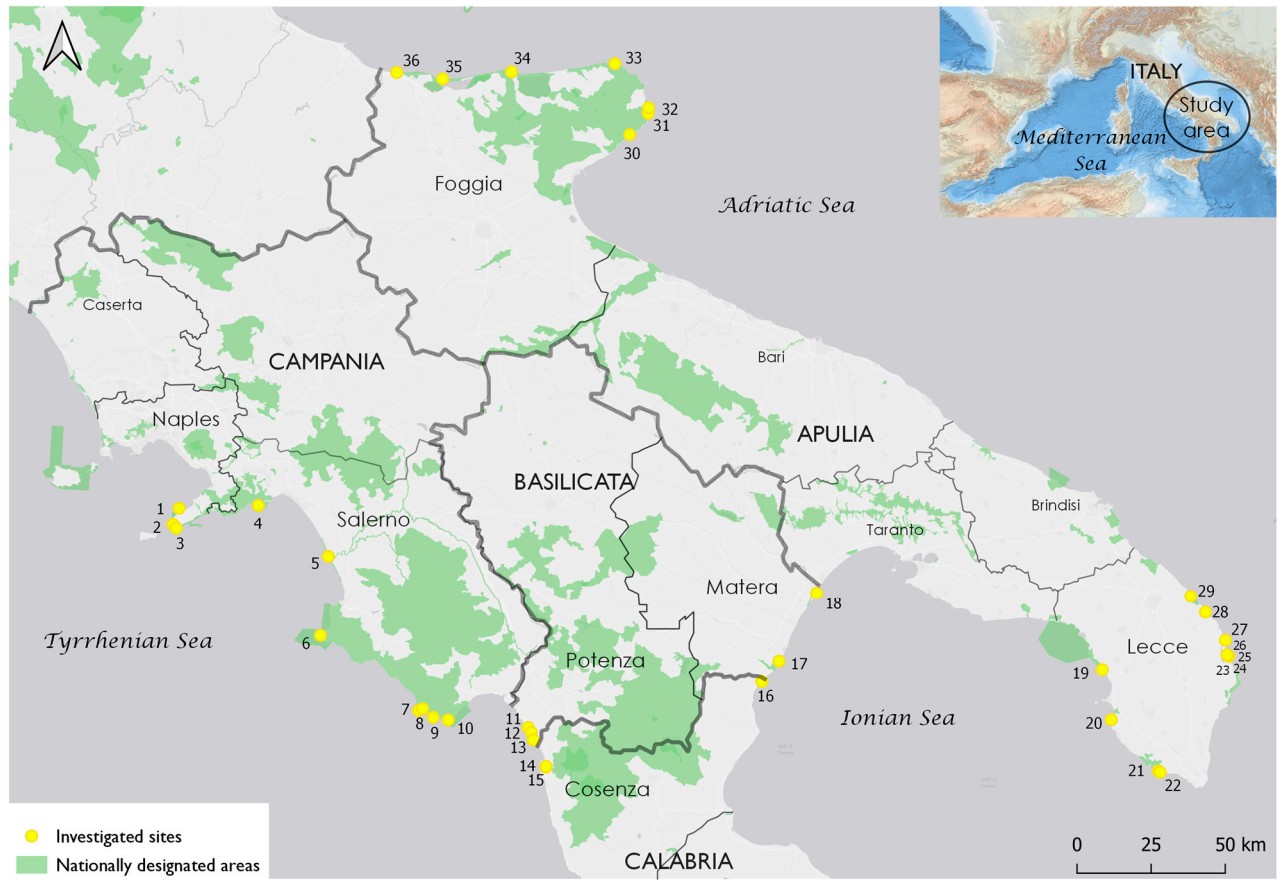

**Figure 1.** Location map of investigated sites.

The provinces of **Naples** and **Salerno** face the Tyrrhenian Sea and belong to the Campania Region that extends for ca. 450 km. Nearly 52% of its total length is characterised by the presence of low sandy beaches mainly located along the alluvial coastal plains of the Volturno, Sele, Sarno, and Alento rivers [18]. These coastal plains originated during the Pliocene and Early Pleistocene as half-graben structures in an extensional tectonic regime induced by the opening of the Tyrrhenian Sea [19]. High rocky coasts characterise the Sorrento Peninsula and the Cilento promontory.

The former, whose structure is represented by the Lattari mountainous ridge, is an elongated WSW–ENE horst that separates two semigrabens, the Gulf of Naples to the north and the Gulf of Salerno to the south, giving rise to a very attractive jagged coast linked to the abrupt topography. This area also has a very high geo-archaeological value due to the presence of well-preserved villas and emerged and submerged maritime structures, such as docks, piers, quays and fishponds, built during the Roman period [20,21]. The Cilento promontory, with ca. 100 km of coastline, marks the border with the region of Basilicata. Due to the presence of a high geological and cultural heritage, firstly, this area was recognized in 1991 as national park and, later, as a biosphere reserve by the United

Nations Educational, Scientific and Cultural Organization (UNESCO). It is noteworthy to state that the entire Campania region has an impressive list of six UNESCO World Heritage Sites—whilst a country such as the United States has 11—and four of them are located near the coast, i.e., Napoli, Pompei, Amalfi and Cilento/Vallo di Diano. To the south, the **Potenza** province (Basilicata) and northwestern **Cosenza** Province (Calabria), have an investigated total length of ca. 40 km (Figure 1). Coastal morphology is mainly characterised by elevated relief with great slope steepness strongly controlled by the Lucania segment of the South Apennine orogeny [22].

To the east, the provinces of **Matera** (Basilicata) and northeastern **Cosenza** (Calabria) face the Ionian Sea along 45 km (Figure 1). The Ionian coastal landscape is essentially characterised by gently seaward-dipping marine terraces, formed during the Pliocene–Pleistocene, gradually descending onto flat coastal plains that are bordered by sandy dissipative beaches limited by shore-parallel dune ridges [23,24].

The provinces of **Foggia** and **Lecce**, with a total length of 512 km, belong to the Apulia region that have a coastline ca. 900 km long facing both the Adriatic and the Ionian seas (Figure 1). The total coast length of Apulia corresponds to 12% of the Italian littoral, and essentially consists of sandy beaches that are ca. 650 km in length. From a geographic viewpoint, the region is characterised by three main domains almost entirely observed within the Mesozoic Apulia carbonate platform [25]: the Gargano promontory (in the north); the Murge highlands (in the central part); and the Serre Salentine (in the south). From a geological/geomorphological perspective, the whole region stands out by hosting 440 identified geosites (especially located along the coast) of very high interest for conservation and mapped as part of the "Geosites Project", partially funded by the European Union [26,27]. In this paper, the provinces of Bari, Taranto and Brindisi were not investigated because they have a high level of coastal human development with respect to Lecce and Foggia; as an example, the latter is almost fully protected under the Gargano National Park, the largest park in Apulia [28].

Moreover, the regions of Campania, Apulia and, to a lesser extent, Basilicata, are very popular 3S tourism destinations. In 2019, Campania and Apulia were ranked among the eight most visited Italian regions before Sicily or Sardinia with, respectively, >22 and 15.40 M visitors [29]. Even if tourists visiting Campania (and particularly Naples) are firstly interested in "Cultural aspects" (38%) ahead of "Seaside tourism" (28%) [30], both regions clearly showed the highest "Tourism Intensity/Density Rates" of coastal municipalities [31]. As an illustration, every year the famous UNESCO Amalfi Coast, which is almost 50 km in length, receives nearly 5 M visitors. Another curious case is the Sorrento Peninsula, which in 2018 received about 2.7 M visitors and >90% were foreign tourists [29]. In the case of Apulia, 49% of the Blue Economy companies were directly linked to beach tourism, which contributed to 47% of blue economy jobs [32]. Most of the touristic activities (and urban areas) are generally concentrated along coastal plains that also have a remarkable value for agricultural and livestock farms in their inner sectors [33]. It must also be said that the southern regions are the most affected by erosion processes in Italy. In fact, Campania, Basilicata and Apulia show similar coastal erosion rates that affect ca. 55% (>60% for Calabria) of their total coastline length, which is indeed a critical challenge to be faced [9].

Finally, and following the criteria set up by Williams [12], sites were located in "Remote" areas (>66%), "Rural" areas (5) and seven were considered as "Resort" beaches; nearly all, 83% of the total amount, were located within protected areas totally or partially covered by different designations applied at regional, national, European or international level such as the Amalfi coast, declared as World Heritage Site (UNESCO) (Table 1). Remote sites frequently required at least a 20 min walk and sometimes were only accessible by sea, e.g., D'I Vranne. Places such as Buondormire (Salerno) or Baia di Campi (Lecce) were considered as "Resort" since only clients from the hotel/resort could have direct access to the beach (the second case is further discussed). Beaches with a high presence of concessions but freely accessible by land through a walk > 300 m from the nearest car parking, were categorized as "Resort" and "Remote", e.g., Scorzone (point 11, Figure 1

and Table 1); this is quite common in Italy but very unusual in other countries. Lastly, in some cases, the designation Libera (i.e., "free", Table 1) was used when it only refers to the free/public sector without beach concessions, e.g., La Secca Libera (Salerno). General site characteristics, i.e., location (province, municipality), beach typology, presence of protected areas, etc., are presented in Table 1.

**Table 1.** General sites characteristics with corresponding location map numbers: province (Pr.), municipality (Mun.), beach typology, coastal length, protected areas and Coastal Scenic Evaluation System (CSES) results, i.e., "D" value and Class, see Methods for explanation.

| Site | Pr. | Mun. | Typology * | Length (m) ** | Protected Areas | D | Class |
|---|---|---|---|---|---|---|---|
| 1. Bagni Regina | Naples | Sorrento | Remote | 10 | Sito Archeologico Villa di Pollio Felice (Cultural Heritage protection) Area Naturale Marina Protetta Punta Campanella SAC & SPA *** | 0.87 | I |
| 2. Cala di Mitigliano | | Massa Lubrense | Remote | 260 | Area Naturale Marina Protetta Punta Campanella SAC & SPA | 0.94 | I |
| 3. Baia di Ieranto | | Massa Lubrense | Remote | 26 | Donated to Fondo Ambiente Italiano (FAI) Area Naturale Marina Protetta Punta Campanella SAC & SPA | 1.00 | I |
| 4. Cavallo Morto | | Maiori | Remote | 152 | UNESCO World Heritage Site Amalfi Coast Parco Regionale dei Monti Lattari SAC & SPA | 0.93 | I |
| 5. Sele Tanagro | Salerno | Eboli | Remote | 1090 | Riserva Naturale Foce Sele-Tanagro Area Protetta Dunale Legambiente Silaris Military zone SAC | 0.72 | I |
| 6. Punta Licosa | | Castellabate | Rural | 190 | Parco Nazionale del Cilento e Vallo di Diano Area Marina Protetta Santa Maria di Castellabate SAC & SPA | 0.80 | II |
| 7. Buon Dormire | | Centola | Resort | 83 | Parco Nazionale del Cilento e Vallo di Diano SAC & SPA | 0.89 | I |
| 8. Arco di Palinuro | | Centola | Rural | 98 | Parco Nazionale del Cilento e Vallo di Diano (zone A2) SAC & SPA | 0.79 | II |
| 9. Cala d'Arconte | | Camerota | Resort | 240 | Parco Nazionale del Cilento e Vallo di Diano (B1) SAC & SPA (only marine) | 0.85 | I |
| 10. Pozzallo | | Camerota | Remote | 65 | Parco Nazionale del Cilento e Vallo di Diano (A2) Area Marina Protetta Costa degli Infreschi e della Masseta SAC & SPA | 1.06 | I |
| 11. D'I Vranne | Potenza | Maratea | Remote | 122 | SAC | 1.17 | I |
| 12. Cala Ficarra | | Maratea | Resort | 32 | SAC | 0.83 | II |
| 13. La Secca Libera | | Maratea | Remote | 205 | SAC | 0.83 | II |
| 14. Scorzone | Cosenza | San Nicola Arcella | Resort & Remote | 330 | No land protection features SAC (only Marine) | 1.00 | I |
| 15. Arco Magno | | San Nicola Arcella | Remote | 29 | No land protection features SAC (only Marine) | 1.03 | I |
| 16. San Nicola Rocca Imperiale | | Rocca Imperiale | Remote | 1850 | No protection features | 0.83 | II |
| 17. Oasi Bosco Pantano | Matera | Policoro | Remote | 1360 | Riserva Naturale Orientata Bosco Pantano di Policoro SAC & SPA (only southern sector) | 0.97 | I |
| 18. Metaponto Libera | | Bernalda | Remote | 890 | Riserva Naturale Metaponto SAC | 0.95 | I |

**Table 1.** *Cont.*

| Site | Pr. | Mun. | Typology * | Length (m) ** | Protected Areas | D | Class |
|---|---|---|---|---|---|---|---|
| 19. Porto Selvaggio | | Nardò | Remote | 35 | Parco Naturale Regionale Porto Selvaggio e Palude del Capitano<br>SAC | 1.02 | I |
| 20. Punta Pizzo | | Gallipoli | Resort & Remote | 350 | Parco Naturale Regionale Isola di Sant'Andrea e litorale di Punta Pizzo<br>SAC & SPA | 0.70 | II |
| 21 Calette di Salve | | Salve | Remote | 72 | No protection features | 1.07 | I |
| 22. Fanciulla | | Salve | Rural | 170 | No protection features | 0.92 | I |
| 23. Toraiello | | Otranto | Remote | 16 | SAC | 0.90 | I |
| 24. Turchi Shore Platform | Lecce | Otranto | Remote | 240 | SAC | 1.09 | I |
| 25. Cala dei Turchi | | Otranto | Remote | 73 | SAC | 0.85 | I |
| 26. Baia dei Turchi | | Otranto | Resort & Remote | 185 | SAC | 0.78 | II |
| 27. San Giorgio | | Otranto | Remote | 210 | SAC | 0.99 | I |
| 28. Torre Specchia | | Melendugno | Rural | 145 | No protection features | 0.87 | I |
| 29. Cesine | | Vernole | Remote | 850 | Riserva Naturale Le Cesine (border)<br>RAMSAR; SAC & SPA | 0.47 | III |
| 30. Fontana delle Rose | | Mattinata | Remote | 215 | Parco Nazionale del Gargano<br>Riserva Naturale Monte Barone<br>SPA | 1.09 | I |
| 31. Portogreco | | Vieste | Remote | 118 | Parco Nazionale del Gargano<br>SAC & SPA | 1.12 | I |
| 32. Baia di Campi | | Vieste | Resort | 540 | Parco Nazionale del Gargano<br>SAC | 0.83 | II |
| 33. Cala del Turco | Foggia | Peschici | Remote | 58 | Parco Nazionale del Gargano<br>SAC | 1.03 | I |
| 34. Torre Calarossa | | Cagnano Varano | Rural | 445 | Parco Nazionale del Gargano | 0.64 | III |
| 35. Morella | | Lesina | Remote | 4180 | Parco Nazionale del Gargano<br>SAC & SPA | 0.56 | III |
| 36. Pineta Marinelle | | Chiuti & Serracapriola | Remote | 3220 | Parco Nazionale del Gargano<br>SAC | 0.53 | III |

* Typologies set up according to Williams [12]; Remote—may be defined by difficulty of access, largely by boat or on foot—a walk of up to 300 m+; Rural—found outside the urban/village environment, not readily accessible by public transport and virtually without facilities—perhaps a small summer shop, car park and/or toilet; Resort—usually located on a beach adjacent to an accommodation/concession complex, and where management is the responsibility of the complex. ** Length is considered when constant scenic conditions are observed. *** Natura 2000—Special Area of Conservation (SAC) and Special Protection Area for Birds (SPA).

## 3. Contextualisation: The Lack of a Coherent National Coastal Strategy

As this paper aims to propose sound management interventions to upgrade/maintain scenic attractiveness of investigated sites, it is of paramount importance to understand the legal framework and tools that regulate beach uses and management in the study area. In Italy, management of coastal areas is quite complex and characterised by a lack of integrated governance and national strategy. Coastal zones cannot be properly regulated because of overlapping laws/regulations at national, regional and municipal levels, e.g., maritime public domain laws, civil code and navigation code, landscape and urban planning regulations at regional/municipal levels, concession laws of public domain, etc. [34,35]. Symptomatic of this institutional fragmentation is the fact that Italy has not yet ratified the Integrated Coastal Zone Management Protocol and does not have a national maritime strategy [36].

Regulation for coastal protection is part of broader environmental legislation usually pertaining to "landscape" issues [37]. Whilst the civil and navigation codes are well

established and together define the maritime public domain (MPD), the body of laws regarding the MDP's owner and administrative functions has undergone numerous changes over the past six decades, empowering the regional administrations. Nonetheless, following long debates, especially regarding the delegation of competencies to the regions —who owns and who manage maritime goods?—the national government remains the owner of all maritime goods, with the exception of Sicily and, from 2014, Friuli-Venezia Giulia [37].

This "regionalisation" trend started several decades ago. In 1968, with the approval of the electoral law no. 108 of 17 February, the constitution of the regions with ordinary statute began concretely. In 1977, a decree of the president transferred management of tourism and recreational uses in the MPD, which was the responsibility of the state, to the regional governments. This transfer of competences was later clarified through the administrative federalism laws (1997–1998) but, in 2004, the "new Code" (Law 42/2004) marked a significant turn in coastal protection by delegating even more power to the regional governments [37]. Indeed, whilst the code still lists a "protection zone", i.e., an area protected and subjected to restrictions on development (Law 431/85), it enabled regional administrations to regulate/add specific restrictions through landscape planning (Pianificazione paesaggistica in Italian). This latter summed up a list of existing territorial regional plans which names varied from place to place. As a result, at the national level, there are no uniform provisions regarding permitted uses within the "protection zone". Beyond these regional plans, all municipalities must also prepare local land use/urban plans, adding another level of complexity [35,37]. Moreover, since 2001, the constitution assigned most administrative functions of the state to the municipalities, except when these are more adequately exercised by a higher-tier administration in accordance with the "principles of subsidiarity, differentiation and adequacy" (Art.118, Constitutional Law 3 of 2001).

Today, enforcement illegalities ascertainment is present in many municipalities and regions. As a matter of fact, 55,020 illegalities in coastal areas were reported in 2021 alone (one for every 133 m of coast) [38]; >35% of national coastal lands falling within the 300 m strip are urbanised and fewer than half of the Italian beaches with good bathing conditions are freely accessible [9]—beaches are not actually privately owned, but leased under a concession regime. After years of automatic/generalised extensions (Decree 400 of 1993), and unfulfilled commitments toward Europe, i.e., the European Bolkestein Directive, Italian legislators seem to have taken a first significant step towards an organic reform of beach concession with the recent approval of the Competition Law (Law 118/2022). By 2024, concession shall be awarded by a public tender procedure. Lastly, as a brief overview, some of the most relevant coastal legislations (cited above) were listed below with their original translation in Italian:

(i) The 1968 Law "Provisions for the election of Regional councils" (Legge n. 108/68 *Norme per la elezione dei Consigli regionali delle Regioni a statuto normale* in Italian),

(ii) Decree no. 616 of 1977 "Transfer of administrative functions to the Regions" (*Decreto n. 616/1977 Soppressione di uffici centrali e periferici delle amministrazioni statali*),

(iii) The 1982 Law "Provisions for the Defence of the Sea" (*Legge* n. 979/82 *Disposizioni per la difesa del mare*),

(iv) The 1985 Law "Urgent provisions for the protection of areas of particular interest" (*Legge n. 431/85 Disposizioni urgenti per la tutela delle zone di particolare interesse ambientale* called *Legge Galasso*), which introduced a "protection zone" (*zona tutelata*) as the area within 300 m from the shoreline,

(v) The 1991 Law for "Environmentally protected areas" (*Legge n. 394/91 Quadro sulle Aree Protette*),

(vi) Decree no. 400 of 1993 "Provisions to determine fees for MPD concessions" (*Decreto n. 400/1993 Disposizioni per la disciplina delle concessioni demaniali marittime e la determinazione dei canoni*),

(vii) The 2004 Law "The Code on Cultural Heritage and Landscape" (*Legge n. 42/2004 Codice dei beni culturali e del paesaggio*),

(viii) The 2022 Law "Annual Market and Competition Law 2021" (*Legge n. 118/2022 Annuale per il mercato e la concorrenza 2021*).

## 4. Methods

### 4.1. Methods Used

The Coastal Scenic Evaluation System (CSES) method was used to assess sites' scenic quality. It was an outcome of an investigation financed by the British Council that was later rewritten and published by Ergin et al. [15,16] and Ergin [17]. The technique is semi-quantitative and provides an excellent synthesis from all previous attempts, e.g., Fines [39], Penning-Rowsell [40,41] and the Country Council for Wales [42,43], knitting together the expert and general public viewpoints plus strong mathematic tools, i.e., fuzzy logic.

In a first step, as a result of a long process of literature review, consultations with local experts and public enquires to a wide spectrum of beach users in Malta, United Kingdom (UK), Turkey and Croatia (>1000 surveys), 26 scenic parameters were selected [15]: 18 physical components—cliff (height, slope, features), beach face (width, colour, type), rocky shore (slope, extent, roughness), dunes, valley, landform, tides, coastal landscape features, vistas, water colour and clarity, vegetation cover and debris; and 8 human parameters—noise disturbance, litter, sewage evidence, built and non-built environment, access type and utilities (Table A1).

In a second step, to compute the weight of each selected parameter—since it was quite obvious that some of the 26 parameters were more relevant than others—new surveys (>500) were carried out with beach users who were asked to rate them from top to bottom [15]. The attractive water colour/clarity as well as the presence of coastal landscape features (see example in Table A1), and the absence of litter, noise, buildings and utilities were the most appreciated parameters. Furthermore, each parameter was ranked from low (1), i.e., absence/poor quality, to high (5), i.e., excellent/outstanding quality.

In a third step, a mathematical model, based on fuzzy logic [44] that enables a system to be looked at as a spectrum rather than black/white or yes/no, was developed by a group of experts. It partially allows for the reduction in possible errors made by the scenic value assessor marking the wrong attribute box in the 26 parameters checklist. As an example, cliff height can be absent (rated 1), present a height between 5 and 30 m, 30–60 m, 60–90 m or >90 m (rated 5) (Table A1); if the scenic evaluator ranks cliff height between 30–60 m, the fuzzy logic gives a score of 1 to such option and a 0.3 score to the two closest options, i.e., 5–30 and 60–90 m cliff-height intervals.

After giving a score to each parameter, an evaluation index (D) is obtained, which enables the classification of sites into five distinct classes: Class I (extremely attractive natural sites, D ≥ 0.85); Class II (natural/attractive sites with high landscape value, 0.65 ≤ D < 0.85); Class III (natural/average sites with medium landscape value, 0.40 ≤ D < 0.65); Class IV (poor sites with medium landscape value and light development, 0.00 ≤ D < 0.40); and Class V (very unattractive, intensively developed urban sites, D < 0.00). Pragmatically, the higher the "D" value is, the better the site's scenery is. Additionally, the software used incorporates a graphical presentation of data, i.e., histograms, weighted averages and membership degree curves, which allow immediate visualization of the state of scores obtained at natural and human parameters (Figure A1). The above is a rapid overview to present the most relevant aspects of the method used but a very detailed description of concepts, mathematical background, etc., can be found in Ergin [17]. Lastly, CSES is very useful in quantifying how scenic quality may be improved when parameters' attributes change (positively or negatively).

### 4.2. Site Selection

In the quest of identifying top scenic sites not (or slightly) impacted by human activities, the present approach was adopted following the methodology applied in Bulgaria, Spain or France [45,46]. In this paper, even if emphasis was placed on Class I and Class II

sites, a few Class III sites (4) were chosen, and field assessed as they all showed high physical value but were significantly affected by beach litter—which is not an irreversible problem.

In a preliminary step, to preselect coastal areas that appeared of great scenic value according to the 26 parameter checklist, a first approximation was achieved via satellite images and land cover viewers, e.g., Google Earth and Copernicus. High-resolution images with good accuracy (nearly 0.25 m) were easily available making it possible to get a first idea about physical parameters such as "Beach face", "Shore platform", "Dunes", "Vegetation cover" and anthropologic aspects such as "Built and Non-Built environment" or "Access type" (Table A1).

A second step mainly consisted of obtaining information and, particularly, images from a ground perspective when doubt arose concerning certain preselected areas, i.e., grey literature, existing papers, official tourism and/or protected areas websites, e.g., the Web Portal of Italian Parks [47]. It must be stressed that a time limit related to access difficulty was set to 90 min (from the nearest car parking by walking). It should be noted that the greatest spatial density of sites can be observed along an heterogeneous coastline in the municipality of Otranto in Lecce (points 23–27, Figure 1), which exhibits a wide variety of sceneries in <15 km of coastal length, i.e., cliffs, dunes, bays, pocket beaches and shore platforms, whilst the opposite was true for homogeneous coastlines such as Matera Province (points 17, 18, Figure 1) or the northern coast of Foggia (34–36, Figure 1). Sectors were selected irrespective of whether they were (or were not) situated within protected areas. After these two steps, a total of 45 sites was preselected.

The last and most important part of the research was obviously the field work as selected areas must be assessed in situ to check/verify previous observations and fill out the CSES checklist; each one of the 26 parameters were scored according to the attribute scale (from 1 to 5; Table A1) in a mid-beach position over sectors 400–500 m in length. Surveys were carried out during normal weather conditions, i.e., when stable conditions ruled, e.g., a storm may alter sand and water colour (points 6 and 16, Table A1). After the field visits, only 36 sites (out of 45) were finally selected and presented in this paper. For example, the southern remote sector of Cesine (Lecce) was visited but finally not chosen because too high of a scenic impact was caused by a set of continuous breakwaters, situated 15 m apart from the shoreline.

## 5. Results

### 5.1. Evaluation Index (D) Values and Classes

Site scores for the evaluation index (D) are presented in Figure 2. Very attractive places such as D'I Vranne (D = 1.17), Portogreco (1.12) or Fontana delle Rose (1.09) stand out from the rest because of their high index values reflected by excellent scores for both physical and human parameters. In total, 24 sites correspond to Class I, i.e., extremely attractive sites, 8 to Class II and 4 to Class III. The pocket pebble beach of Pozzallo (Salerno) is a good illustration of Class I. Located in the core area of the Cilento e Vallo di Diano National Park (zoning 1 "very restrictive"), a long walk up to 35 min, is required to reach the beach. For Class II sites, at places, low "D" values were associated with low scoring for physical parameters, e.g., San Nicola Roca Imperiale (0.83; Cosenza), however, often they were directly linked to high visual impacts of human influence. Sele-Tanagro (0.72) is an interesting example of a second-class site. Good ratings were essentially obtained for "Beach" characteristics (points 4–6, Table A1), "Dunes", "Vistas" and "Vegetation" (points 10, 15 and 18, Table A1) but the large amounts of beach litter and the continuous accumulation of vegetation debris (>50 cm, point 20, Table A1) present along the dune front, considerably downgraded its scenic quality. Lastly, Cesine (0.47) is a good case of a Class III site, for which good scores were noted for "Beach characteristics", "Dunes", "Vistas" and "Water colour/clarity", however, "Litter" obtained the lowest grade (1) because of the continuous accumulations recorded. Site scores (CSES) are reported in Table A2. These three examples of Class I, II and III were presented in Figure A1.

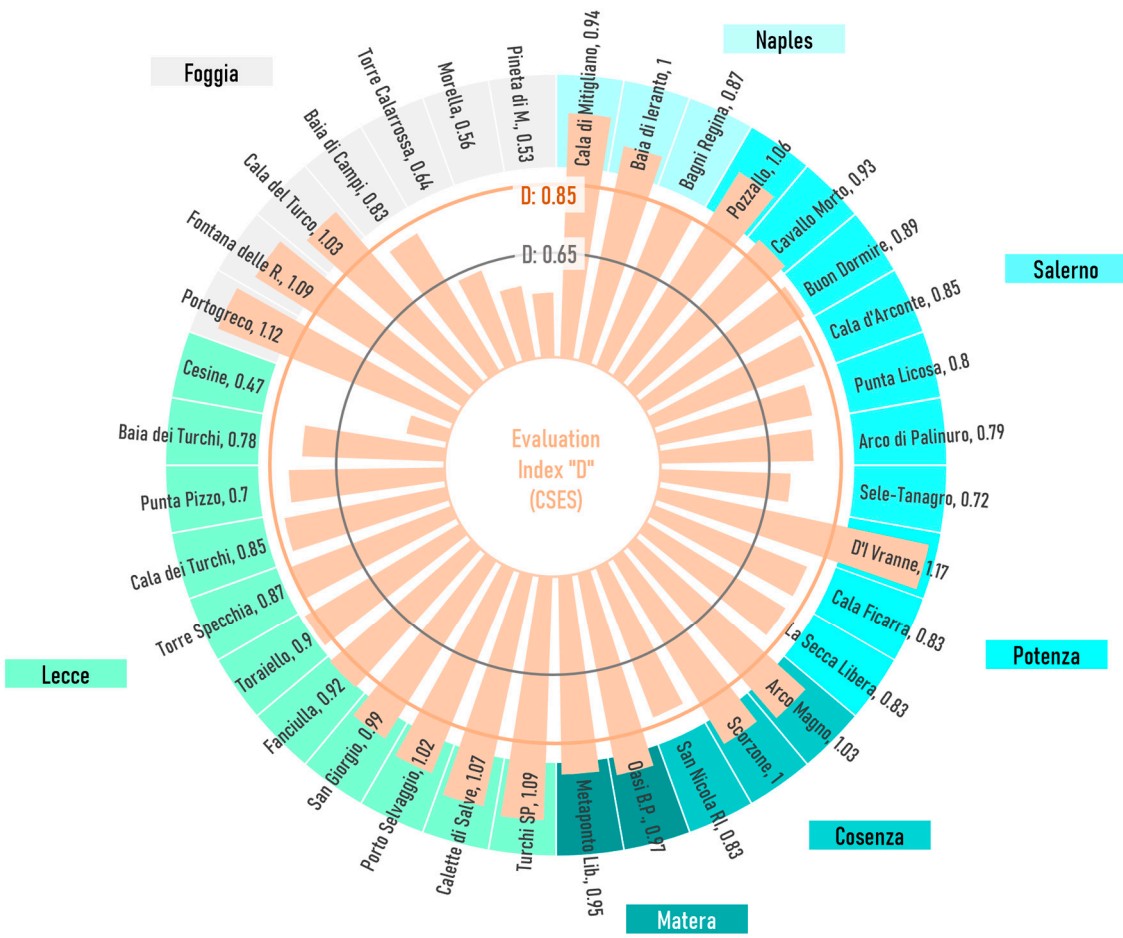

**Figure 2.** Site scores for evaluation index (D) presented according to the province to which they belong to, and limits of Class I (D ≥ 0.85), Class II (0.65 ≤ D < 0.85) and Class III (0.40 ≤ D < 0.65).

## 5.2. Physical Parameters

### The Tyrrhenian Coast

This area comprises 15 sites stretching from the Sorrento Peninsula to the northern province of Cosenza (Figure 1). The Campania region is considered to have one of the most diversified coastal systems in Italy, with alternating high/low coastal sectors and exceptional geomorphological settings linked to volcanic-tectonic processes [48,49]. In the Sorrento Peninsula, this was particularly reflected by top scores for "Cliff" parameters (points 1–3, Table A2) where the abrupt topography slopping down to the sea, gives rise to very attractive pocket pebble beaches, e.g., Baia di Ieranto and Cavallo Morto (Figure 3A). South of the Campania region, the large mountainous promontory of Cilento, interposed between the coastal plains of Policastro and the Salerno gulfs, favours the presence of steep cliffs, rocky headlands, small sized beach, impressive karstic formations and numerous islets close to the shore, reflected in "Cliff" and "Coastal landscape features" scores (points 1–3 and 14, Table A2), e.g., Arco di Palinuro, particularly famous for its eponymous natural arch shaped in a Jurassic limestone. At Punta Licosa, the large shore platform dating to the Middle Pleistocene gives rise to good scores for "Rocky shore" parameters (points 6–9, Table A2).

To the north, the Sele river plain displays a completely different coastal scenery with sand beach (from the Holocene) surrounded by herbaceous dunes, pines forest and low landform. The hilly and mountainous coastal relief of the Apennines, in the Basilicata and northern Calabria, also favours high ratings for "Landform" (point 14, Table A2), and particularly at sites with a far vista opened background such as La Secca Libera and Scorzone.

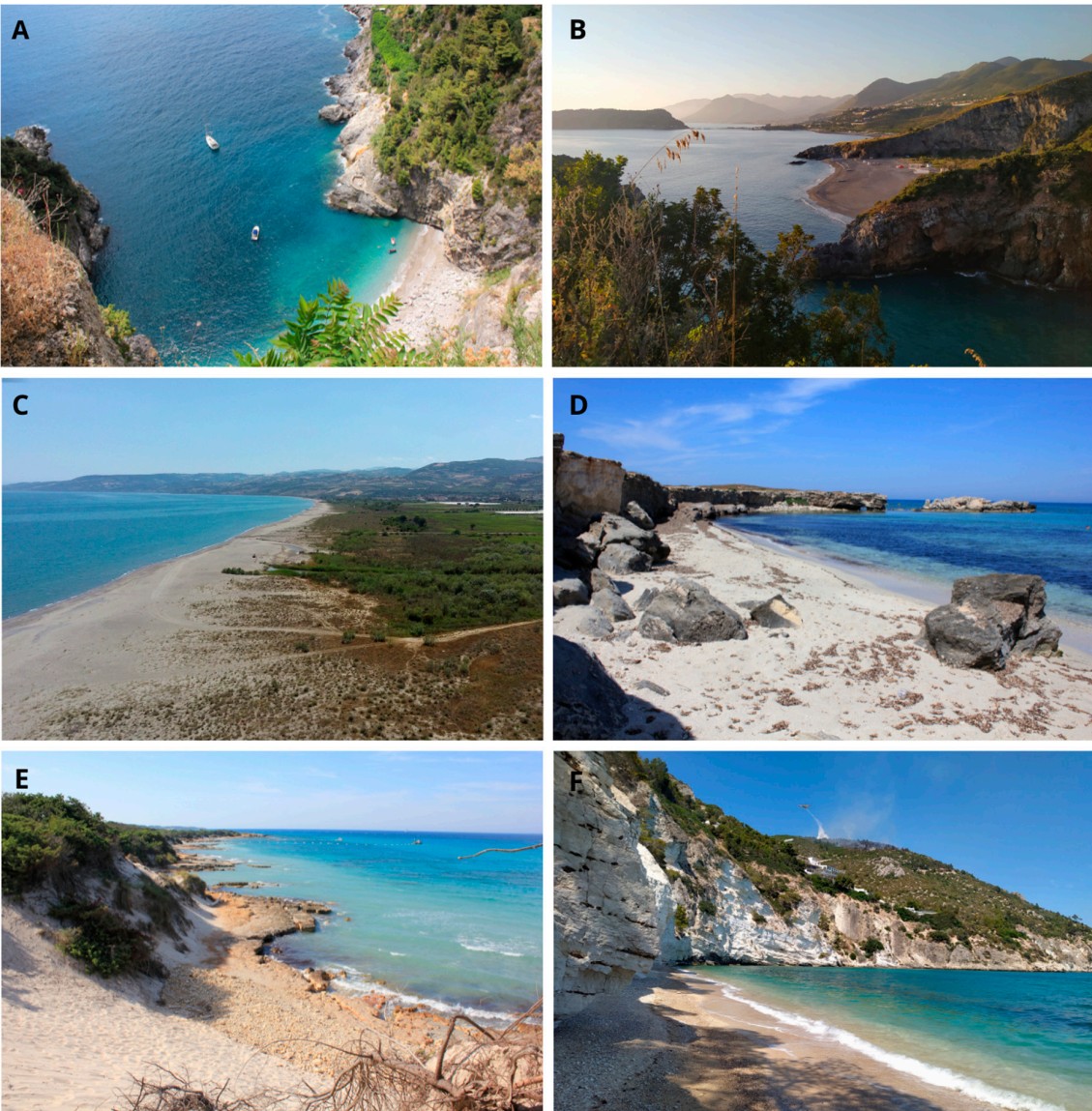

**Figure 3.** Pocket beach of Cavallo Morto (D: 0.93, Class I) at Sorrento peninsula, Amalfi coast, with a cliff height > 100 m (**A**); coastal relief and landscape features along the northern coast of Cosenza with Scorzone in the background (D: 1.00, Class I; (**B**)); eastern coast of Cosenza stretching the Ionian Sea (D: 0.83, Class II; (**C**)); crystal-turquoise waters and islet at Torre Specchia (D: 0.87, Class I; (**D**)); San Giorgio's dunes and shore platform (D: 0.99; Class I) in Lecce Province (**E**); Fontanelle di Rose (D: 1.09, Class I) at Gargano National Park, only accessible by sea, with aerial firefighting in the background (**F**).

This southern part is particularly marked by the beautiful contrast of colours between the green vegetation and the dark/black sand from volcanic deposits—considered within the parameter "Landscape features" for its singularity—all bordered by turquoise clear water, e.g., D'I Vranne. It must be stressed that most of the Tyrrhenian sites showed very clear water (most of them rated 5, point 16, Table A2), which frequently depends on the volcanic rocks outcropping in the water shield, as well as an exuberant Mediterranean vegetation (point 17, Table A2). This jagged coast also gives rise to caves, stacks, islets, e.g., Scorzone (Figure 3B), and, sometimes, to spectacular places such as Arco Magno; a very enclosed pebble/shale pocket beach well known for its majestic natural arch shaped by wave action.

*The Ionian and Adriatic Coasts*

Whilst the western coast of Basilicata and Cosenza is quite mountainous, the landscape of their respective eastern coast, stretching along the Ionian Sea, is marked by low clay hills and wide valleys favouring good scores for "Vistas", i.e., very open line of sight (point 15, Table A2), and sand/gravel beaches with dunes systems, e.g., San Nicola Imperiale (Figure 3C) or Oasi Bosco Pantano. The latter, located within a Natural Reserve managed by the World Wildlife Fund (WWF), stands out for being one of the last testimonies of lowland forest formation in southern Italy that survived the uncontrolled land drainage and intensive agriculture [50]. These characteristics were particularly reflected by good scores for "Vegetation Cover" and "Landscape features" (Table A2).

To the southeast is the Salento peninsula that essentially corresponds to the Lecce Province with a geological structure made of a thick Mesozoic carbonate sequence covered by Tertiary and Quaternary deposits [51]. All the peninsula landscape was modelled by karst processes favouring very aesthetic sceneries composed of white/gold sandy beaches, calcareous shore platforms, caves, stacks, and dolines, bathed in crystal-turquoise waters, e.g., Torre Specchia (Figure 3D). A curious case is San Giorgio (Lecce) showing a singular eroding dune complex composed of several ridges (>6 m height, rated "5" point 10) emplaced on a calcareous shore platform (Figure 3E). Several interconnected brackish lakes can also be observed along the northern Adriatic coast of Salento, as seen in Cesine.

Lastly, there is the well-known Gargano promontory located along the province of Foggia (northern Apulia). Since 1991, this extensive area has been almost fully protected as a national park (>120,000 ha) for encompassing a wide variety of habitats from ancient woodlands (belonging to the primeval Umbra forest and consisting mainly of beeches) to actual coastal forests of pines and oaks, grasslands, marshes and coastal lagoons. Whilst the northern part is characterised by lowlands and sandy beaches backed by extensive dune systems, pine forests and lakes, e.g., Morella, the promontory coast (emerged from the Adria plate) gives raise to very steep "white" cliffs made up of limestone and dolomite rocks from the Jurassic–Cretaceous periods [52], e.g., Fontanelle di Rose (Figure 3F). Most places recorded high ratings for "Cliff", "Beach" (white/gold sandy beach), "Water colour" (turquoise) and "Vegetation cover" (mainly pine forests). As observed in Salento, the general physiography of this coastline, shaped by karst process, also favours high scores for "Landscape features", i.e., caves, stacks, reefs and islets, headlands, etc.—>4000 sinkholes and 600 caves were registered within the whole Gargano area [52].

*5.3. Human Parameters*

In the following lines the different human parameters are analysed in order to have a general overview on the main impacts of human activities/settlements on coastal scenery:

**"Noise disturbance"** is generally linked to human activities carried out near the beach, e.g., playing loud music (bars), jet skis, heavy traffic (because of a nearby littoral road or a railway), overcrowded scenarios, etc. It was generally low during field work observations (carried out in June). Noisy sites were especially observed at small pocket beaches, such as Arco Magno or Buondormire, and at sites considered as "Resort" because of the higher numbers of visitors and/or nearby lidi, e.g., Cala Arconte, Punta Pizzo or Baia di Campi. Most of these sites were easily accessible by a walk <10 min. Baia dei Turchi, a very famous beach in Salento, obtained a low score (3) because of loud music (Table A2).

**"Beach Litter"** refers to discarded man-made objects. It is a major concern for Italian coasts and, even more, at remote sites (where beaches are free!), which stand out for their general lack of periodic control/cleaning operations demonstrating the low interest of local managers for such areas. Even if most sites were characterised by "few scattered items" (rated 4), essentially alluding to macro litter, numerous investigated sectors showed "single accumulations" (rated 3), "full strand line" (rated 2) and/or "continuous accumulations" (rated 1), critically lowering their scenic attractiveness (Figure 4). This was particularly observed over kilometres of coastlines along the northern Foggia beaches located within Gargano National Park, i.e., Torre Calarossa (rated 1), Morella (rated 2), and

Pineta Marinelle (rated 2). A similar critical case was observed at Cesine Natural Reserve (Lecce), where continuous accumulations were lying at dune toes (Figure 4). An additional sector at Cesine was field visited and also characterised by a full strand line of litter—and finally not chosen because of its low scenic quality (case described in Methods). Items were, in general, mainly stranded by sea currents and composed of plastic items (bottles, bags and cups), glass drinks bottles, fast-food packaging, foamed polystyrene, cans and fishing waste. Along the Tyrrhenian coast, at Sele-Tanagro (Regional Natural Reserve), large amounts of litter were noted, probably linked to the eponymous nearby river, and at La Secca Libera (rated 2), in the province of Salerno. It must be said that good scores were observed at investigated resort beaches, e.g., Arconte, Baia dei Turchi and Baia di Campi.

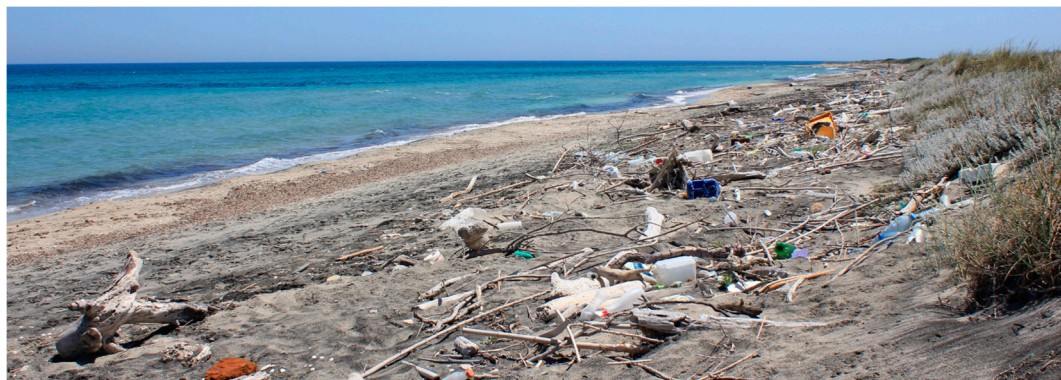

**Figure 4.** Continuous accumulations of beach litter at Cesine Natural Reserve, Lecce Province.

**"Sewage"** was virtually absent at investigated sectors. It is usually visible in urban or village beach typologies but hardly expected in remote or rural areas.

**"Non-built environment"** was essentially constituted by natural trees, forests, etc., allowing top scores (5). Only three sites obtained a rate of "4", namely, Cavallo Morto, Torre Calarossa and Baia di Campi. In the case of the two first places, it was linked to the presence of agriculture fields, i.e., pasture fields for Calarossa and terrace farming for the second (very common along the Amalfi coast), whilst Baia di Campi was backed by a camping resort.

**"Built-environment"** showed excellent scores since most sites were located in remote areas surrounded by completely natural environments. Sites such as Cala Arconte, Cala Ficarra and Baia di Campi (resorts) obtained a lower rate because of restaurants/bars built near the beach that cannot be removed. A curious case is Baia di Ieranto, a very remote pocket beach accessible by a 45-min walk, where a yellow building, recently reformed, stands out near the bay (rated 4). This coastal land was donated in 1987 to the Fondo per l'Ambiente Italiano (equivalent to the National Thust in UK) by Italsider, one of the largest Italian steel industrial groups. A limestone quarry was located in the bay since the early 1900s and the building, before used for carry works, is today a museum [53]. Lastly, the defensive coastal tower rising from the promontory of Porto Selvaggio (since 1569), known as Torre dell'Alto, obtained a top grade for "historical environment".

**"Access type"** is linked to the visual impact of roads and/or car park areas—good scores were generally observed ($\geq 4$). At Fanciulla, a low score (3) was due to vehicles that illegally crossed the dunes to get as close as possible to the beach. This bad practice particularly raises concerns about beach users' and dune protection in addition to the scenic impact. The absence of a buffer zone at Torre Specchia (rated 3) makes the car parking very visible from the beach, considerably affecting its scenic quality. Most investigated resort beaches were rated "4" since parking areas were usually perceptible, i.e., Arconte, Ficarra and Baia di Campi.

**"Skyline"** usually corresponds to silhouettes of buildings (or others human settlements) not in harmony with the environment, e.g., villages, cities, factories, industrial ports, etc. At most sites, sensitively designed housing in nearby villages were appreciable

(rated 4), which can be linked to the investigated beach typologies. Only two sites showed lower ratings (3): Scorzone and Punta Pizzo. The former was backed by a very high traffic bridge (at 1 km distance), whilst, for the second, it was linked to the town of Gallipoli and the excessive presence of recreational boats anchored in nearby coastal waters (an illegal practise very common in this area) obstructing the complete view of the bay.

"**Utilities**" parameter covers a large list of human items such as power lines, telephone lines/poles, lighting, communication, groins, breakwaters, lifeguard towers, sewage outfall, railways, etc. Nonetheless, low/poor scores observed at most sites (Table A2) were basically associated with intrusive structures devoted to recreational/seasonal use, i.e., beach bars, beach umbrellas, sunbeds, etc. (Figure 5A–D). It must be recalled that sites' characteristics investigated in this paper, mainly located in natural/remote environments, do not reflect the Italian typicity (Figure 5B), i.e., 43% of the countrys' beaches are under private concession. Even in remote areas, it is not easy to find a "free beach" without a lido and its associated recreational services. Several places were preselected, and field visited, but finally, not chosen because of excessive utilities and associated problems (overcrowded scenarios, noise disturbance, amongst others). This was the case of Torre dell' Orso (Lecce), famous for being in front of the "two sister stacks" (Due Sorelle in Italian), which was visited but not tested giving that >450 umbrellas and >900 sunbeds were counted over 150 m of beach (that was 50 m in width). Other sites such Cala d'Arconte, Cala Ficarra, Scorzone, Baia dei Turchi and Baia di Campi (Figure 5A) were field tested and chosen despite their poor scoring for "Utilities", as they all displayed outstanding values at physical and remaining human aspects. Lastly, poor scoring at Arco di Palinuro (rated 2) was due to the presence of several support structures aiming to protect beach users from recurrent rockfall particularly affecting the arch (nearly 20 m in height) [54].

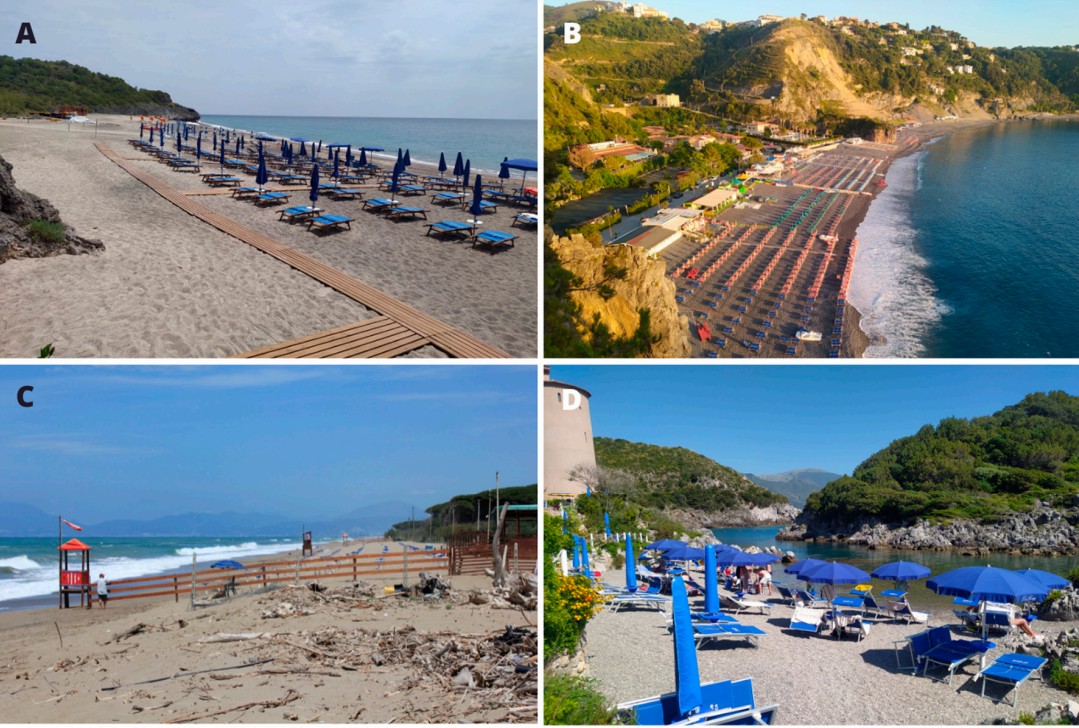

**Figure 5.** Scenic impact of beach facilities at Arconte (Gargano National Park) with >250 sunbeds along 200 m of beach length (**A**); typical beach case observed along the Tyrrhenian coast near Arco Magno in Cosenza (**B**); beach concession, which does not respect the free 5 m areas along the shoreline, within the Sele-Tanagro Natural Reserve at the border with the military restricted zone (**C**); pocket beach at Punta La Secca fully occupied by recreational facilities (**D**).

Last, taking into account the results obtained in the investigated regions concerning human parameters, it is possible to state that the downgrading of most sites is due to the presence of beach litter and recreational facilities and other issues related to "private" beaches and this is due to the clear lack of coordination/integrated management actions (discussed in the next section). Several judicious interventions are proposed in Table 2 with the aim of demonstrating how sites could recover their natural beauty and, at the same time, to quantify how both issues can adversely affect sceneries. For example, if periodic cleaning operations would be carried out at Class III sites, e.g., to obtain a rate of "4", they would upgrade their scenic class, e.g., Cesine (0.47). Additionally, if most second-class sites would reduce the scenic impact of intrusive utilities, their aesthetic value would drastically increase leading most of them to Class I (Table 2). Regarding "Access type", Torre Specchia could improve its "D" value by mitigating the visual impact of parking areas as well as Fanciulla by controlling the illegal access to dunes (Table 2).

**Table 2.** Proposal of judicious interventions to increase sites attractiveness and upgrade of associated evaluation index (D).

| Site | Parameter and Related Present Score | Interventions * | "D" Value | |
|------|-------------------------------------|-----------------|-----------|---|
| | | | Present | Post-Intervention |
| Cala di Mitigliano | Litter (3) | Establish periodical cleaning operations | 0.94 | 1.08 |
| Sele-Tanagro | Litter (3) | | 0.72 | 0.87 |
| La Secca Libera | Litter (2) | | 0.83 | 1.13 |
| Cala dei Turchi | Litter (3) | | 0.85 | 1.02 |
| Cesine | Litter (1) | | 0.47 | 0.77 |
| Torre Calarossa | Litter (1) | | 0.64 | 0.91 |
| Fanciulla | Access type (3) | Limit the access of vehicles | 0.92 | 0.96 |
| Torre Specchia | Access type (3) | Establish a narrow buffer zone between beach and car parking | 0.87 | 0.91 |
| Punta Pizzo | Skyline (3) | Reduce/regulate anchoring of recreational boats in nearshore waters | 0.82 | 0.96 |
| Arco di Palinuro ** | Utilities (2) | Reduce visual impact of cliff protection structures | 0.79 | 0.94 |
| Cala d'Arconte | Utilities (1) | Allow a very limited number of utilities sensitively designed to be in harmony with the nature | 0.85 | 1.07 |
| Cala Ficarra | Utilities (1) | | 0.83 | 1.04 |
| Scorzone | Utilities (2) | | 1.00 | 1.14 |
| Baia dei Turchi | Utilities (1) | | 0.78 | 1.08 |
| Baia di Campi | Utilities (1) | | 0.83 | 1.07 |

* Interventions allowing a significant improvement of the considered parameter (to reach a score of 4). ** This case is quite complex as "safety" prevails over "scenery", that said, without such evident protection structures, Palinuro would be a top scenic site.

## 6. Discussion

Beach management is indeed a complex task that demands a holistic view of beach functions, i.e., protection, conservation and recreation. Nonetheless, a common mistake lies in considering beaches as a whole rather considering them by typologies. In the same manner that urban/village sites must be easily accessible and well-equipped, in remote and rural areas, scenic quality that is among the five parameters ("Big Five") of the greatest significance to coastal visitors should be prioritized [12]. Based on this, it is not rational that some remote sites showed full equipment services associated with hundreds of sunbeds, umbrellas, beach bars, etc., deteriorating the very essence and positive image of these natural sceneries. Of course, such a dilemma is not only characteristic to Italian beaches

but is seen globally at 3S destinations where conflicts usually arise between recreational activities (and associated short-term benefits) and long-term collective interests. The overlapping of Italian laws/regulations, linked to the institutional fragmentation (local, regional and national levels), has led to chaotic conditions to properly manage the country's coastal areas. Lastly, it is also unacceptable that rural and remote public beaches are generally characterised by large amounts of beach litter whilst beaches under concessions are usually clean. Thereby, to better understand how complex and particular the Italian case is, the following sections aim to present a more detailed analysis of beach concessions and litter issues by answering a set of interrogations that arose from the fieldwork.

### 6.1. An Analysis of Beach Concessions

The topic of beach concession is generally quite complex, but the Italian situation is surely unprecedented. As previously stated, only half of the sandy coastlines is freely accessible and usable for bathing, since nearly 43% is occupied by beach concessions and 7.2% is not accessible due to the low microbiological quality of coastal waters essentially linked to poor sewage systems [55]. As a result, at many places, it is observed that the presence of *Escherichia coli* and/or *Enterococcus* spp. exceed legal EU limits. In the last three years, the number of concessions has increased by 12.5%, from 10.812 (2018) to 12.166 (2021) [56]. In some investigated municipalities, up to 70% of beaches were under concessions, e.g., San Nicola Arcella in Cosenza (73%) or Peschici in Foggia (74%) [9]. All the above may lead to the following concerns:

#### Occupation and Access Regulations

To date, there is no national legislation establishing a maximum percentage of beaches that may be granted in concession. Since the "Bassanini law" and the constitutional reform (in 2001) with the decentralisation of competencies at regional and municipal levels, functions related to the granting of concessions were assigned to the municipalities. Some regions have intervened by setting limits, but few have implemented really incisive measures. Among the virtuous legislative cases Apulia can be cited. By the regional Law 7/2006, it requires at least 60% of the MPD available length (within each municipality) to be reserved for public use and free bathing. However, in the practise, most of municipalities keep adopting their municipal plans without considering these limits, and the region is currently engaged in numerous legal dispute proceedings. Ridiculous restrictions for a minimum length of public sectors were set up in the Calabria (30%) and Campania (20%) regions [9]. To give a counter-example, the French national legislation establishes a minimum of 80% of the coastal length (and beach surface) that must remain free of any structure, equipment, installation, etc. (Environmental Code).

Another critical concern is access regulation. Whilst access to beaches must be free of charge and, in theory, a right for all citizens stipulated by the civil and navigation codes, it is too often denied in certain regions. This is the case of many municipalities in Campania, where widespread illegality does not allow users to freely access the beach and/or walk along the shoreline because of gates or diverse obstacles installed to reserve beach areas for customers of beach clubs, hotels, etc. Legally, everyone has the right of vertical/horizontal access, and particularly within the 5 m from the shoreline. This was surprisingly experienced during the field work. In Salerno, access was denied to "reach" (stay and assess) Punta La Secca, a pebble pocket beach situated in a sensitive designed environment, backed by a Chapel and few old buildings, with impressive views on the Scoglio U'Tuppu islet. As shown in Figure 5D, the whole beach surface was occupied by sunbeds and beach umbrellas—nearby, La Secca Libera was assessed showing general excellent scores but large amounts of litter. Another curious and unacceptable case is Baia di Campi in Vieste (Foggia Province). To freely access the beach, users must walk on a steep and dangerous (and unmarked) path along the nearby cliff, while clients from the camping resort have direct access from the nearby parking area.

*Disparities between Concession Leases and Price for Users*

The first aspect to highlight is the questionable absence of updated and detailed data on the fees paid for concessions for using the MPD. In recent years (2016–2020), revenues were approximately estimated in ca. EUR 97 million per year [57]. Reports carried out by different Italian national journals also show that most beach concessions paid between 1000 and 5000 EUR per year, and report the total absence of data for many concessions/municipalities—map viewers of concession leases and fees are available online [58,59]. For example, no information regarding this aspect was found concerning the coast of Maratea in Potenza (>20 concessions and 3 investigated sites). Some clubs pay higher amounts, between 5000 and 10,000 and a very few >10,000 EUR per year. It is quite curious to see in various investigated municipalities such as Mattinata (Foggia), numerous concessions paying less than EUR 1000 (13 out of 35). In this regard, it must be pointed out that the disparity between derisory fees paid and prices of sunbeds and beach umbrellas, is constantly increasing in all regions. As an example, in Palinuro (Salerno) the price for a single sunbed with umbrella, in the front rows, during August is around 170 EUR/week, whilst at some places like Gallipoli (Lecce), it reaches 282 EUR/week [60].

*Protected Areas Versus Beach Concessions*

Another striking issue concerns the usual presence of beach clubs within protected areas, which are supposed to have a high level of protection, under the guidelines for natural reserves or national parks (zone A2-B1), e.g., Foce Sele-Tanagro Natural Reserve (regional), Metaponto Reserve Natural (national) and Cilento e Vallo di Diano National Park—even if, obviously, occupation is quite lesser than in unprotected areas. At Sele-Tanagro, although the chosen sector for field assessment was characterised by good scores for "Utilities" because it is backed by a military zone (along a coastal sector 1 km in length), several beach concessions remained along the Natural Reserve (16 km in length). Another worrying aspect is the fact that concessions were clearly not sensitively designed and/or hidden in the back dunes but are rather lying on the beach front with a capacity of hundreds of sunbeds and beach umbrellas, and, in some cases, do not respect the free 5 m areas along the shoreline (Figure 5C). As previously mentioned, Cala Arconte is another curious case. Located within the Cilento e Vallo di Diano National Park in "zone B1", supposed to be a "restricted use" area, the whole beach (200 m in length) was occupied by >250 sunbeds, umbrellas and other beach facilities (Figure 5A). While many small hotels lie in harmony/hidden amongst its foliage with unobtrusive vista—a perfect example of sensitive tourism—these facilities considerably downgraded its landscape quality (Table 2). It is not a unique case but rather an illustration of what happened at several sites within this area, e.g., at the twin beach of Arconte, and elsewhere, along the investigated regions. To give more examples, at the entrance of Metaponto Natural Reserve, an extensive beach club also remained on the shoreline with, once again, hundreds of sunbeds, etc., just as well as at Punta Pizzo, situated within a Regional Natural Park (zone B1). This latter obtained intermediate scores (3) for "Utilities", since the scenic impact of facilities was quite visible from the Libera sector, whilst the former was characterised by top grades as the free beach without structures was >1 km in length. Once more, at Gargano National Park, a few beach clubs spread out within "zone 1", which are presumed to have the most restrictive approach regarding human activities.

These observations evince that most of the Italian protected areas are very (or too) permissive with respect to the presence of concessions, which show full facilities within zones that are theoretically devoted to prioritise environmental values, including landscape beauty. Such situations very unlikely to happen in national/natural parks or natural reserves of neighbour 3S countries such as France or Spain—some good examples of landscape management within protected areas in Spain can be found in Mooser et al. [45,61]. It is obvious that concessions can be a powerful mean of providing higher quality services, filling the mismanagement of the state/regions/municipalities, but a delicate balance must be found between sustaining scenic beauty/integrity and economic development. Equilibrium also needs to be achieved between primary versus secondary services in coastal

areas of great scenic value. At specific places, lifeguard stations are certainly indispensable because of rip currents, but beach clubs associated with sunbeds, umbrellas, etc., should be reduced and/or moved away from the beach. To give an example, at Ses Salines Natural Park in Formentera (Spain), well-designed beach bars are authorised in the back dune area close to the parking, this way not affecting the beach scenic quality [45]. That said, good practices were observed at Porto Selvaggio in Lecce (Regional Natural Park; D = 1.02, Class I), with the use of only primary services, e.g., information panels, litter bins, both wisely installed, and control of access to avoid/reduce overcrowded scenarios mainly due to the small beach dimensions. Another interesting case is Pozzallo (Salerno) where a beach bar lies hidden amongst the foliage without altering scenic quality.

### 6.2. An Analysis of Beach Litter

Another critical concern is beach litter [11,62]. Indeed, a recent report from the NGO Legambiente, carried out along 53 Italian beaches (belonging to 14 regions), identified an average of 834 abandoned waste items every 100 m [63], whilst the threshold value (established at the European level) to consider a beach in good environmental status is <20 litter items/100 m. Other surveys performed within the framework of the Interreg Med ACT4LITTER project (carried out in 26 protected areas and 11 countries), with the aim to carry out a snapshot assessment of marine litter at coastal and marine protected areas in 11 Mediterranean countries, showed that the Italian protected beaches were among the most contaminated (five Italian beaches were in the top six) [64], findings also supported by Francesco et al. [65]. Fortibuoni et al. [66] also pointed out that the Adriatic coast ranked as the most polluted area, in term of beach macro-litter, followed by the Western Mediterranean Sea and the Ionian/Central Mediterranean areas; this raises the following points.

### Leaning of Public and Private Beaches

Italy does not have a national waste management plan for beach and marine litter, but rather has delegated this function to regional administrations who, in turn, redelegated to municipalities. As an example, in the Apulia region, an Ordinanza Balneare (ordinance for beach management issues) is approved yearly by the region to regulate the exercise of municipal activities falling within the maritime public domain, i.e., recreational use, bathers' safety, cleaning operations, access, facilities, amongst others, with specific regulations for public and private beaches [67]. It particularly states that all coastal municipalities are entrusted to ensure hygiene, cleanliness and waste recollection at public beaches whilst, at private ones, this duty is assigned to the respective concessions (art. 6).

### Situation at Protected Areas

The framework on protected areas, i.e., Law 394/1991, lays down the fundamental principles for the establishment/management of Italian protected natural areas, classifying them into various levels of protection: national parks; regional and interregional nature parks; and nature reserves (state or regional); amongst others. Whereas regional protection features are run by the various regional administrations, mainly through territorial coordination plans (or park plans), national parks come under the auspices of the Italian Ministry of Environment with the requirement to establish a park plan (prevailing over regional plans). In both cases, zoning plans must be established to decide the limits of acceptable use and development for each zone, including their control and maintenance. To give an example, the park plan of the Cilento National Park prohibits carrying out of mechanical beach cleaning operations within the "zone A1" (an area of "strict nature reserve"). This was the case for Pozzallo where only a few scattered items were identified. In zones C–D (lower environmental value), traditional cleaning/maintenance operations are usually conducted by municipalities. At Cala Arconte (zone B1), litter is collected by several concessions curiously authorized to be there. The case of the Gargano National Park is quite symptomatic of the procedural complexity for Italian national parks to settle a "Plan Park" (>50% do not have a park plan) [68]. In line with Fortibuoni et al. [66], results concerning beach quality obtained from this paper present the poorest scoring along

the Apulia region (on the Adriatic coast) and particularly at sites within the Gargano National Park and the Cesine Natural Reserve, i.e., Torre Calarossa, Morella, Pineta Marinelle (Foggia) and Cesine (Lecce). This coastal sector is already well-known for accumulating marine debris due to its geographic location; in this area the predominant marine currents (flowing from the northern Adriatic) are stopped by the Gargano promontory, favouring litter deposition. Many observed litter items are linked to the frequent, nearby presence of fishing and aquaculture activities. According to Legambiente [69], this situation, which is common along several kilometres of coastline, is due to the absence of an integrated beach-litter management plan.

*General Considerations: Beach Litter Versus Scenic Quality*

This paper clearly stands out how littering can critically affect sites scenic beauty (see evaluation index "D" in Table 2). A place like Torre Calarossa could jump two scenic classes, from Class III to Class I, solely by solving litter problems. As observed in others countries [45,46,61], the absence of periodic cleaning operations along coastal remote sites can be linked to the difficult access for clean-up machines and, in some cases, to the specific regulations of certain protected areas, e.g., zone "A", where only manual collection is authorized. However, the huge number of items in certain areas and the fact that most of them have been lying on the beach for a long time, constitute clear evidence of the low interest of competent authorities/managers. As an example, it is interesting to highlight that in Menorca (Balearic Islands), cleaning operations are, in some cases, carried out with boats to reach very remote sites [45,70]. Building on the recommendations from several national NGOs, e.g., WWF and Legambiente [38], there is an urgent need to strengthen/improve/make more efficient the control of competent authorities and regional environmental protection agencies—both under the supervision of the ISPRA (Italian Institute for Environmental Protection and Research). An effective legal mechanism should be also designed to financially sanction administrations responsible for mismanagement, such as the local municipalities. As a possible recommendation, it could be interesting to reproduce at remote/rural sites the French Trait Bleu program with the project Bac à Marée (tidal bins in English) [71], aiming to encourage and promote eco-citizen initiative to collect beach litter stranded by sea currents. These wooden bins (with low scenic impact) are installed by local managers/municipalities and filled-up by beach users.

Further, it could be wise to exploit the CSES method as a baseline to label/recognise sites of outstanding landscape quality, e.g., Class I sites (D ≥ 0.85). Thereby, a "scenic award" would imply, amongst other benefits, a follow-up and therefore the need to establish a periodical clean-up because of regular inspections to check sites environmental and scenic quality. Arrangements could be formed between protected area managers and respective municipalities. In addition, such arrangements could be focused on other management aspects such as access regulations, beach facilities (linked with scenic quality), as a mean to recognize good practises from municipalities and/or protected areas—EU and/or national NGO criteria may be used.

## 7. Conclusions

Coastal landscapes have become a natural/economic resource to be challenged, and scenic assessment constitutes a mandatory issue as it provides a practical basis for managers to establish sound management strategies for reaching long-term goals. However, in the present context, the case of Italy is of greatest concern. Indeed, the evolution of beach concessions in the last few years (+12.5%), the lack of general control and uniform provisions to properly manage coastal areas, and the unnumbered illegalities committed despite existing laws in force, have put the remaining free beaches of great scenic value in an endangered and delicate situation in the near future.

Herein, in the quest of identifying top scenic sites barely affected by human activities, 36 sites (from 7 provinces and 4 southern regions) were selected, field visited and assessed using the CSES method. Despite the above context, 24 sites fall within Class I, i.e., were extremely attractive (D ≥ 0.85) because of their exceptional geomorphological settings,

water colour/clarity and high vegetation covers (amongst others). It was demonstrated that beach litter and intrusive recreational facilities were the factors that had the greatest impact on sites' scenic quality. Several judicious interventions were proposed with the aim of demonstrating how sites could recover their natural beauty and, in the same time, to quantify how both issues can adversely affect sceneries. Indeed, all Class III sites (4) could be upgraded only by cleaning operations, whilst most of second-class sites (8) could jump to Class I by reducing intrusive (not essential) beach facilities.

That said, in practise, these interventions are unlikely to be carried out. Nonetheless, the results obtained could be wisely used as a baseline for the establishment of a novelty "coastal scenic label", bringing multiple benefits to local communities, protected areas and beach users, and more interest in scenery than beach services. It could also bring considerable positive side effects, e.g., periodical inspections, which could be of great interest for tackling beach concessions and litter problems. It must also be stressed that beach concession and scenery preservation can be compatible when activities are sensitively calibrated to be in harmony with the natural environment. Finally, the findings achieved (and the method used) could be of great use for national NGOs, such as Legambiente, that are very active in monitoring and reporting the mentioned issues. For example, such proposals would fit within the new Italian Coastal Landscape Observatory scope (Osservatorio Paesaggi Costieri Italiani), recently created/designed and, particularly, focused on the transformation of coastal sceneries and sustainable practises.

**Author Contributions:** A.M. and G.A. designed the study and participated in all phases. E.P. and A.R. provided specific information related to the study area, i.e., historic background, tourism trends, geomorphological aspects and legislation, amongst others. P.P.C.A. made contributions regarding the conceptual approach. A.M. and G.A. carried out the field work observations in June 2022 and E.P. participated in the evaluation of Lecce Province. All authors have read and agreed to the published version of the manuscript.

**Funding:** This research was partially funded by Università degli Studi di Napoli Parthenope (Italy) as the first author was supported by a PhD scholarship under the program "Environmental Phenomena and Risk", cycle 35th.

**Acknowledgments:** Thanks to Gabriele Lami from the National Research Group for Coastal Environment issues (GNRAC, Italy) for reviewing the legal context. This work is a contribution to the Andalusia Research Group RNM-328 (University of Cádiz, Spain).

**Conflicts of Interest:** The authors declare no conflict of interest.

## Appendix A

**Table A1.** Checklist of the 26 physical/human parameters considered by the Coastal Scenery Evaluation System (CSES) method; applied in >40 countries.

| No. | Physical Parameters | | Weight * | Rating | | | | |
|-----|---------------------|---|----------|--------|---|---|---|---|
| | | | | 1 | 2 | 3 | 4 | 5 |
| 1 | | Height (m) | 0.02 | Absent | $5 \leq H < 30$ | $30 \leq H < 60$ | $60 \leq H < 90$ | $H \geq 90$ |
| 2 | CLIFF | Slope | 0.02 | < 45° | 45–60° | 60–75° | 75–85° | circa vertical |
| 3 | | Features ** | 0.03 | Absent | 1 | 2 | 3 | Many (>3) |
| 4 | | Type | 0.03 | Absent | Mud | Cobble/Boulder | Pebble/Gravel | Sand |
| 5 | BEACH FACE | Width (m) | 0.03 | Absent | $W < 5$ or $W > 100$ | $5 \leq W < 25$ | $25 \leq W < 50$ | $50 \leq W \leq 100$ |

**Table A1.** *Cont.*

| No. | Physical Parameters | | Weight * | Rating | | | | |
|---|---|---|---|---|---|---|---|---|
| | | | | 1 | 2 | 3 | 4 | 5 |
| 6 | | Color | 0.02 | Absent | Dark | Dark tan | Light tan/bleached | White/gold |
| 7 | ROCKY SHORE | Slope | 0.01 | Absent | <5° | 5–10° | 10–20° | 20–45° |
| 8 | | Extent | 0.01 | Absent | <5 m | 5–10 m | 10–20 m | >20 m |
| 9 | | Roughness | 0.02 | Absent | Distinctly jagged | Deeply pitted and/or irregular | Shallow pitted | Smooth |
| 10 | DUNES | | 0.04 | Absent | Remnants | Fore-dune | Secondary ridge | Several |
| 11 | VALLEY | | 0.08 | Absent | Dry valley | (<1 m) Stream | (1–4 m) Stream | River/limestone gorge |
| 12 | SKYLINE LANDFORM | | 0.08 | Not visible | Flat | Undulating | Highly undulating | Mountainous |
| 13 | TIDES | | 0.04 | Macro (>4 m) | | Meso (2–4 m) | | Micro (<2 m) |
| 14 | COASTAL LANDSCAPE FEATURES *** | | 0.12 | None | 1 | 2 | 3 | >3 |
| 15 | VISTAS | | 0.09 | Open on one side | Open on two sides | | Open on three sides | Open on four sides |
| 16 | WATER COLOR and CLARITY | | 0.14 | Muddy brown/grey | Milky blue/green | Green/grey/blue | Clear/dark blue | Very clear turquoise |
| 17 | NATURAL VEGETATION COVER | | 0.12 | Bare (<10% vegetation) | Scrub/garigue (marram, gorse) | Wetlands/meadow | Coppices, maquis (±mature trees) | Variety of mature trees |
| 18 | VEGETATION DEBRIS | | 0.09 | Continuous (>50 cm high) | Full strand line | Single accumulation | Few scattered items | None |
| | Human Parameters | | | | | | | |
| 19 | NOISE DISTURBANCE | | 0.14 | Intolerable | Tolerable | | Little | None |
| 20 | LITTER | | 0.15 | Continuous accumulations | Full strand line | Single accumulation | Few scattered items | Virtually absent |
| 21 | SEWAGE DISCHARGE EVIDENCE | | 0.15 | Sewage evidence | | Same evidence (1–3 items) | | No evidence of sewage |
| 22 | NON-BUILT ENVIRONMENT | | 0.06 | None | | Hedgerow/terracing/monoculture | | mixed cultivation ± trees/natural |
| 23 | BUILT ENVIRONMENT | | 0.14 | Heavy Industry | Heavy tourism and/or urban | Light tourism and/or urban | Sensitive tourism and/or urban | Historic and/or none |
| 24 | ACCESS TYPE | | 0.09 | No buffer zone/heavy traffic | No buffer zone/light traffic | | Parking lot visible from coastal area | Parking lot not visible from coastal area |
| 25 | SKYLINE | | 0.14 | Very unattractive | | Sensitively designed high/low | Very sensitively designed | Natural/historic features |
| 26 | UTILITIES **** | | 0.14 | >3 | 3 | 2 | 1 | None |

* Obtained from cross-cultural evaluation via public surveys (>500) in Turkey, UK, Malta and Croatia [15]. ** Cliff special features: indentation, banding, folding, screes and irregular profile. *** Coastal landscape features: Peninsulas, rock ridges, irregular headlands, arches, windows, caves, waterfalls, deltas, lagoons, islands, stacks, estuaries, reefs, fauna, embayment, tombola, etc. **** Utilities: power lines, pipelines, street lamps, groins, seawalls, revetments, restaurants, etc.

**Table A2.** Site scores obtained from CSES parameters: physical (1–18) and human aspects (19–26).

| Parameter | | (1) Bagni Regina | (2) Cala di Mitigliano | (3) Baia di Ieranto | (4) Cavallo Morto | (5) Sele-Tanagro | (6) Punta Licosa | (7) Buon Dormire | (8) Arco di Palinuro | (9) Cala d'Arconte | (10) Pozzallo | (11) D'I Vranne | (12) Cala Ficarra | (13) La Secca Libera | (14) Scorzone | (15) Arco Magno | (16) San Nicola Imperiale | (17) Oasi Bosco Pantano | (18) Metaponto Libera | (19) Porto Selvaggio | (20) Punta Pizzo | (21) Calette di Salve | (22) Fanciulla | (23) Toraiello | (24) PR Tuchi | (25) Cala dei Turchi | (26) Baia dei Turchi | (27) San Giorgio | (28) Torre Specchia | (29) Cesine | (30) Fontana delle Rose | (32) Baia di Campi | (33) Cala del Turco | (34) Torre Calarossa | (35) Morella | (36) Pineta Marinelle |
|---|---|---|---|---|---|---|---|---|---|---|---|---|---|---|---|---|---|---|---|---|---|---|---|---|---|---|---|---|---|---|---|---|---|---|---|---|
| 1–3 Cliff | Height | 2 | 4 | 3 | 5 | 1 | 1 | 4 | 5 | 2 | 3 | 3 | 2 | 1 | 4 | 3 | 1 | 1 | 1 | 1 | 1 | 1 | 1 | 2 | 2 | 2 | 1 | 2 | 1 | 4 | 4 | 2 | 1 | 1 | 1 | 1 |
| | Slope | 5 | 5 | 3 | 5 | 1 | 1 | 5 | 5 | 4 | 5 | 5 | 4 | 1 | 5 | 5 | 1 | 1 | 1 | 1 | 1 | 1 | 1 | 5 | 5 | 5 | 1 | 5 | 1 | 5 | 4 | 5 | 1 | 1 | 1 | 1 |
| | Features | 4 | 5 | 4 | 4 | 1 | 1 | 5 | 5 | 3 | 5 | 5 | 3 | 1 | 4 | 5 | 1 | 1 | 1 | 1 | 1 | 1 | 1 | 3 | 3 | 3 | 1 | 3 | 1 | 4 | 5 | 4 | 1 | 1 | 1 | 1 |
| 4–6 Beach face | Type | 3 | 3 | 4 | 4 | 5 | 3 | 4 | 4 | 5 | 4 | 4 | 4 | 3 | 4 | 4 | 4 | 4 | 5 | 4 | 5 | 5 | 5 | 5 | 1 | 5 | 5 | 4 | 5 | 5 | 3 | 4 | 4 | 5 | 5 | 5 |
| | Width | 2 | 3 | 3 | 3 | 3 | 3 | 3 | 5 | 5 | 5 | 3 | 4 | 3 | 5 | 4 | 5 | 5 | 5 | 3 | 5 | 3 | 4 | 2 | 1 | 3 | 4 | 3 | 3 | 3 | 3 | 3 | 3 | 4 | 4 | 3 |
| | Colour | 3 | 4 | 4 | 3 | 3 | 3 | 4 | 3 | 3 | 4 | 3 | 3 | 4 | 3 | 3 | 3 | 3 | 4 | 5 | 4 | 5 | 5 | 5 | 1 | 5 | 5 | 5 | 5 | 3 | 5 | 5 | 5 | 4 | 5 | 5 |
| 7–9 Rocky shore | Slope | 1 | 1 | 1 | 1 | 1 | 3 | 1 | 1 | 1 | 1 | 1 | 1 | 3 | 1 | 1 | 1 | 1 | 1 | 5 | 1 | 2 | 2 | 2 | 2 | 1 | 1 | 2 | 1 | 1 | 1 | 1 | 1 | 2 | 1 | 1 |
| | Extent | 1 | 1 | 1 | 1 | 1 | 5 | 1 | 1 | 1 | 1 | 1 | 1 | 5 | 1 | 1 | 1 | 1 | 1 | 5 | 1 | 4 | 3 | 5 | 5 | 1 | 1 | 4 | 1 | 1 | 1 | 1 | 1 | 4 | 1 | 1 |
| | Rough. | 1 | 1 | 1 | 1 | 1 | 4 | 1 | 1 | 1 | 1 | 1 | 1 | 3 | 1 | 1 | 1 | 1 | 1 | 5 | 1 | 3 | 3 | 5 | 5 | 1 | 1 | 5 | 1 | 1 | 1 | 1 | 1 | 5 | 1 | 1 |
| 10. Dunes | | 1 | 1 | 1 | 1 | 3 | 1 | 1 | 1 | 1 | 1 | 1 | 1 | 1 | 1 | 1 | 1 | 1 | 3 | 1 | 5 | 3 | 3 | 1 | 1 | 1 | 1 | 5 | 1 | 3 | 1 | 1 | 1 | 1 | 1 | 5 |
| 11. Valley | | 1 | 1 | 1 | 1 | 1 | 1 | 1 | 1 | 1 | 1 | 1 | 1 | 1 | 1 | 1 | 1 | 1 | 1 | 1 | 1 | 1 | 1 | 1 | 1 | 1 | 1 | 1 | 1 | 1 | 1 | 1 | 1 | 1 | 1 | 1 |
| 12. Skyline landform | | 1 | 1 | 1 | 1 | 1 | 1 | 1 | 1 | 4 | 4 | 1 | 1 | 4 | 5 | 1 | 3 | 1 | 1 | 1 | 1 | 1 | 1 | 1 | 1 | 1 | 1 | 1 | 1 | 1 | 1 | 1 | 1 | 1 | 3 | 1 |
| 13. Tides | | 5 | 5 | 5 | 5 | 5 | 5 | 5 | 5 | 5 | 5 | 5 | 5 | 5 | 5 | 5 | 5 | 5 | 5 | 5 | 5 | 5 | 5 | 5 | 5 | 5 | 5 | 5 | 5 | 5 | 5 | 5 | 5 | 5 | 5 | 5 |
| 14. Landscape features | | 3 | 5 | 5 | 3 | 3 | 3 | 5 | 5 | 3 | 1 | 5 | 4 | 4 | 5 | 5 | 1 | 1 | 1 | 3 | 3 | 4 | 3 | 3 | 3 | 4 | 3 | 3 | 5 | 3 | 3 | 5 | 5 | 3 | 3 | 1 |
| 15. Vistas | | 1 | 3 | 1 | 1 | 4 | 4 | 1 | 3 | 4 | 1 | 4 | 4 | 4 | 5 | 2 | 5 | 4 | 4 | 3 | 4 | 4 | 5 | 2 | 4 | 3 | 4 | 4 | 4 | 5 | 4 | 3 | 4 | 3 | 4 | 5 |
| 16. Water colour | | 5 | 5 | 5 | 5 | 4 | 5 | 5 | 5 | 5 | 5 | 5 | 5 | 5 | 5 | 5 | 5 | 5 | 5 | 5 | 5 | 5 | 5 | 5 | 5 | 5 | 5 | 5 | 5 | 5 | 5 | 5 | 5 | 5 | 4 | 4 |
| 17. Vegetation cover | | 4 | 4 | 4 | 4 | 5 | 5 | 5 | 3 | 5 | 5 | 5 | 4 | 5 | 4 | 4 | 3 | 4 | 5 | 5 | 4 | 4 | 4 | 5 | 5 | 4 | 5 | 5 | 3 | 3 | 4 | 5 | 5 | 5 | 4 | 4 |
| 18. Vegetation debris | | 5 | 5 | 4 | 4 | 1 | 1 | 1 | 4 | 5 | 5 | 4 | 4 | 1 | 5 | 5 | 4 | 5 | 4 | 5 | 3 | 3 | 3 | 5 | 5 | 3 | 5 | 1 | 3 | 1 | 5 | 4 | 5 | 5 | 1 | 1 |
| 19. Noise disturbance | | 4 | 5 | 5 | 5 | 5 | 5 | 4 | 4 | 4 | 5 | 5 | 4 | 5 | 4 | 4 | 5 | 5 | 4 | 5 | 4 | 5 | 4 | 5 | 5 | 5 | 3 | 5 | 5 | 5 | 5 | 5 | 4 | 5 | 5 | 5 |
| 20. Litter | | 5 | 3 | 4 | 4 | 3 | 4 | 4 | 4 | 5 | 4 | 4 | 5 | 2 | 5 | 5 | 4 | 4 | 4 | 4 | 4 | 4 | 5 | 3 | 5 | 4 | 4 | 1 | 5 | 4 | 5 | 4 | 5 | 1 | 1 | 2 |
| 21. Sewage evidence | | 5 | 5 | 5 | 5 | 5 | 5 | 5 | 5 | 5 | 5 | 5 | 5 | 5 | 5 | 5 | 5 | 5 | 5 | 5 | 5 | 5 | 5 | 5 | 5 | 5 | 5 | 5 | 5 | 5 | 5 | 5 | 5 | 5 | 5 | 5 |
| 22. Non-Built Environment | | 5 | 5 | 5 | 4 | 5 | 5 | 5 | 5 | 5 | 5 | 5 | 5 | 5 | 5 | 5 | 5 | 5 | 5 | 5 | 5 | 5 | 5 | 5 | 5 | 5 | 5 | 5 | 5 | 5 | 5 | 5 | 4 | 4 | 4 | 5 |
| 23. Built environment | | 5 | 5 | 4 | 5 | 5 | 4 | 5 | 5 | 4 | 5 | 5 | 4 | 5 | 5 | 5 | 5 | 5 | 5 | 5 | 5 | 5 | 5 | 5 | 5 | 5 | 5 | 5 | 4 | 5 | 5 | 5 | 4 | 5 | 5 | 5 |
| 24. Access type | | 5 | 5 | 5 | 5 | 5 | 5 | 5 | 5 | 5 | 5 | 5 | 4 | 5 | 5 | 5 | 5 | 5 | 5 | 5 | 5 | 5 | 5 | 3 | 5 | 5 | 5 | 5 | 3 | 5 | 5 | 5 | 4 | 4 | 4 | 5 |
| 25. Skyline | | 5 | 5 | 5 | 4 | 4 | 5 | 4 | 4 | 4 | 5 | 4 | 4 | 5 | 3 | 5 | 4 | 4 | 4 | 5 | 3 | 4 | 4 | 5 | 5 | 5 | 5 | 5 | 4 | 4 | 4 | 5 | 4 | 4 | 5 | 5 |
| 26. Utilities | | 4 | 4 | 4 | 4 | 5 | 4 | 4 | 2 | 1 | 4 | 5 | 1 | 5 | 2 | 5 | 5 | 5 | 5 | 4 | 3 | 5 | 5 | 5 | 5 | 5 | 1 | 4 | 4 | 5 | 5 | 4 | 2 | 5 | 5 | 5 |

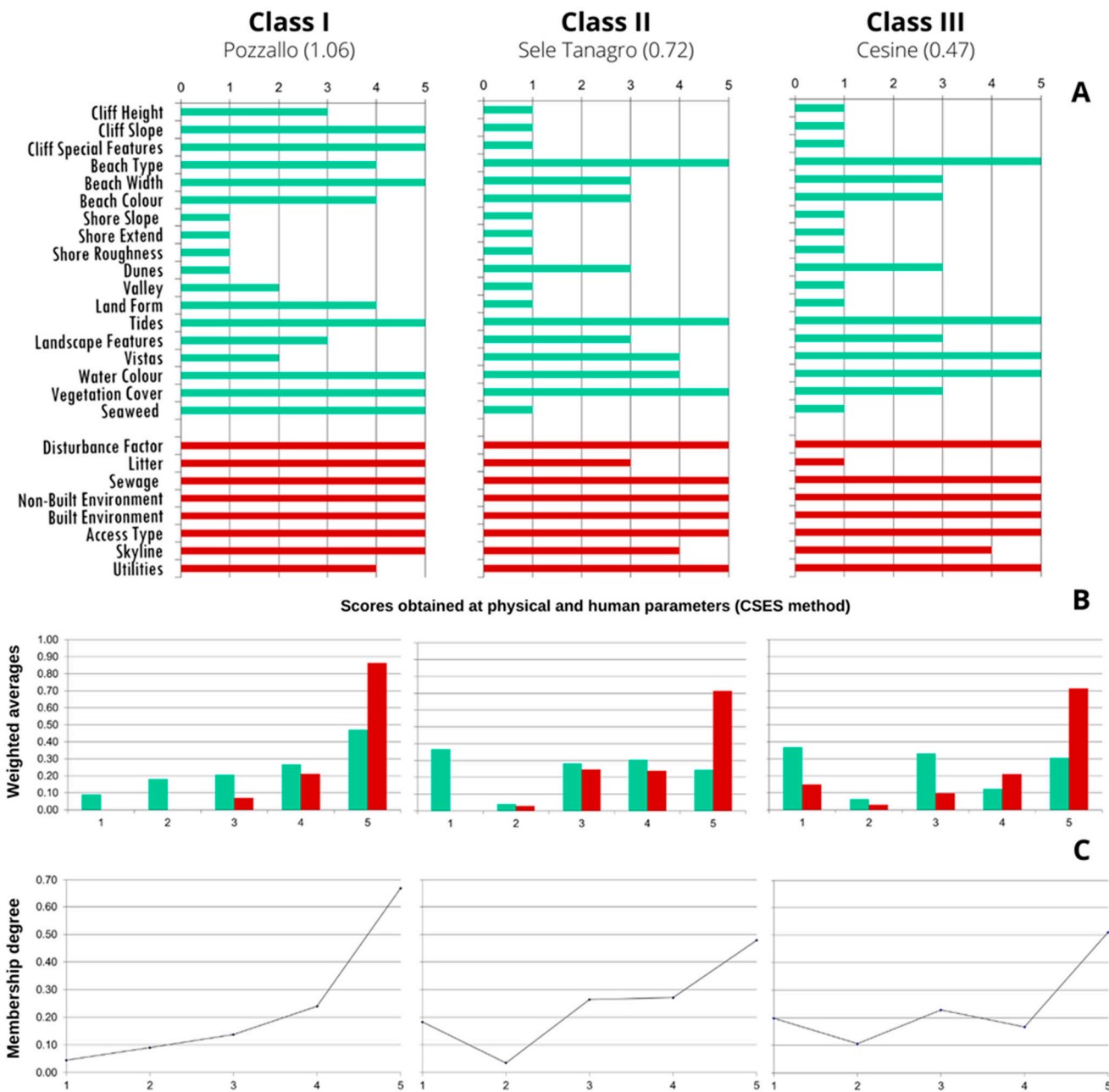

**Figure A1.** Scenic evaluation histograms (**A**), weighted averages (**B**) and membership degree curves (**C**) for Pozzallo (D: 1.06, Class I), Sele-Tanagro (D: 0.72, Class II) and Cesine (D: 0.47, Class III). A look at the weighted average figure instantaneously points out the potential ranking of the scenic evaluation, as well as a membership degree curve rising to the right reflects a high landscape quality due to the low rating on attributes 1 and 2 (and vice versa for a left-skewed curve).

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
