# Peer review of "Beach Scenic Quality versus Beach Concessions: Case Studies from Southern Italy"

_land, doi:10.3390/land12020319_

Round 1

Reviewer 1 Report

Dear Authors,

The article is interesting.

I just have some minor comments on the article.

In general, please make sure that you explain all abbreviations the first time they are mentioned in the manuscript!

It would be nice to read some evaluation of the beaches, e.g. Fig.3. – I assume that C shows an example of lower scenic value but it would be better not to leave it to the readers to judge the photos but add some information about why you show these photos on a particular Figure. Furthermore, all figures and tables should be self-explanatory, in this case, you might say that it is as the caption is telling us what is on the picture but the figure should have a much higher (added) value with more information.

Mentioning the self-explanatory manner of the figures and table, I would also suggest adding the explanation for the abbreviations (see Table 2 where CSES is not explained, so it is not understandable as a stand-alone table).

However, I have major concerns about the Discussion. As you mention at the beginning: "One of the main objectives of this paper is to propose solutions to maintain and/or improve coastal scenic value." I think it is important to do this in the results section, based on your investigations. After you make some proposals for maintenance or improvements, only then can you discuss this issue.

Also, the various overlapping or contradiction of the legislation must be discussed in the result section first.

Otherwise, I think the paper can be fixed easily to make it publishable.

Due to the above-mentioned details, I need to give a major.

Best regards, Reviewer X

Author Response

Dear Reviewer, thank you very much for your feedback. Please, find attached our responses to each comment.

Reviewer 2 Report

Beach Scenic Quality versus Beach Concessions: Case Studies  from Southern Italy

The paper provides a classification of coastal sites of great scenic value in Southern Italy, evaluating direct and indirect human disturbance  and Authors propose management interventions to improve the landscape quality.

The subject matter is very interesting and the results actually provide meaningful and novel insights on the monitoring of beach scenic quality and on the implementation of interventions to improve landscape quality.

However, the text is too long and the number of paragraphs and sub-paragraphs are too many. It would be necessary to move several technical insights (e.g  legal issues, physical parameters, beach concessions) in supplementary materials, and summarize the main results in max 3-4 pages, and the same for the paragraph of Discussion. In addition, results and discussion should be better split.

 Hereafter some other specific comments:

Lines 88-89: “Alarming reports, carried out in various Italian regions, revealed that  were, on average, recorded 10 litter items per square meter, most of them (>80%) consisting of plastic materials”

Lines 762  and the relative sub-paragraph

REV: scientific literature on beach litter in Mediterranean coasts  is missing (e.g.  Fossi, M.C. et al. 2020. Assessing and mitigating the harmful effects of plastic pollution: The collective multi-stakeholder driven Euro-Mediterranean response. Ocean Coast. Manag. 2020, 184, 105005; Battisti, C.;  Poeta, G. et al. 2016. An Unexpected Consequence of Plastic Litter Clean-Up on Beaches: Too Much Sand Might Be Removed. Environ. Pract. 2016, 18, 242–246; de Francesco, M.C. et al. 2019. Natural protected areas as special sentinels of littering on coastal dune vegetation. Sustainability 2019, 11, 5446)

Figure 1 caption: please describe the acronym CCDA

Table 1 caption: please describe the different types mentioned in the column Typology

Line 219 : “the Piano Territoriale Regionale or Piano di Indirizzo Territoriale and the Piano Regionale delle Coste”. Please translate all Italian names along the text and delete the Italian form.

Line 328: “…..via satellite images and land cover viewers”. Please include references

Line 335: “….grey literature, official web sites of tourism, and existing papers”. Please include references

Lines 347-349: “An assessor goes to a site and, in a mid-beach position over sectors 400–500 m in length, looks at each one of the 26 parameters and gives a score to each of them according to the attribute scale (from 1 to 5; Table 2). “  This a very critical point. The assessor’s expertise should be described. Is he/she a geologist? The score attribution may vary a lot according to  the person that makes the assessment. Moreover, the assessor’s evaluations have been validated somehow?

Figure 2: two provinces are missing in the graph

Author Response

(The authors gave the same response as above.)

Round 2

Reviewer 1 Report

Dear Authors,

I think the paper improved enough to be published.

Best regards, Reviewer X

Author Response

Dear Reviewer, thank you!